# Risk factors for mortality in COVID-19 patients in sub-Saharan Africa: A systematic review and meta-analysis

**Ben Bepouka**[1]*, **Nadine Mayasi**[1], **Madone Mandina**[1], **Murielle Longokolo**[1], **Ossam Odio**[1], **Donat Mangala**[1], **Marcel Mbula**[1], **Jean Marie Kayembe**[2], **Hippolyte Situakibanza**[1]

**1** Infectious Diseases Unit, Kinshasa University Hospital, Faculty of Medicine, University of Kinshasa, Kinshasa, Democratic Republic of the Congo, **2** Pneumology Unit, Kinshasa University Hospital, Faculty of Medicine, University of Kinshasa, Kinshasa, Democratic Republic of the Congo

* benbepouka@gmail.com

## Abstract

### Aim

Mortality rates of coronavirus-2019 (COVID-19) disease continue to increase worldwide and in Africa. In this study, we aimed to summarize the available results on the association between sociodemographic, clinical, biological, and comorbidity factors and the risk of mortality due to COVID-19 in sub-Saharan Africa.

### Methods

We followed the PRISMA checklist (S1 Checklist). We searched PubMed, Google Scholar, and European PMC between January 1, 2020, and September 23, 2021. We included observational studies with Subjects had to be laboratory-confirmed COVID-19 patients; had to report risk factors or predictors of mortality in COVID-19 patients, Studies had to be published in English, include multivariate analysis, and be conducted in the sub-Saharan region. Exclusion criteria included case reports, review articles, commentaries, errata, protocols, abstracts, reports, letters to the editor, and repeat studies. The methodological quality of the studies included in this meta-analysis was assessed using the methodological items for non-randomized studies (MINORS). Pooled hazard ratios (HR) or odds ratios (OR) and 95% confidence intervals (CI) were calculated separately to identify mortality risk. In addition, publication bias and subgroup analysis were assessed.

### Results and discussion

Twelve studies with a total of 43598 patients met the inclusion criteria. The outcomes of interest were mortality. The results of the analysis showed that the pooled prevalence of mortality in COVID-19 patients was 4.8%. Older people showed an increased risk of mortality from SARS-Cov-2. The pooled hazard ratio (pHR) and odds ratio (pOR) were 9.01 (95% CI; 6.30–11.71) and 1.04 (95% CI; 1.02–1.06), respectively. A significant association was found between COVID-19 mortality and men (pOR = 1.52; 95% CI 1.04–2). In addition, the risk of mortality in patients hospitalized with COVID-19 infection was strongly influenced by

**Data Availability Statement:** All relevant data are within the manuscript and its Supporting Information files.

**Funding:** The authors received no specific funding for this work.

**Competing interests:** The authors have declared that no competing interests exist.

chronic kidney disease (CKD), hypertension, severe or critical infection on admission, cough, and dyspnea. The major limitations of the present study are that the data in the meta-analysis came mainly from studies that were published, which may lead to publication bias, and that the causal relationship between risk factors and poor outcome in patients with COVID-19 cannot be confirmed because of the inherent limitations of the observational study.

## Conclusions

Advanced age, male sex, CKD, hypertension, severe or critical condition on admission, cough, and dyspnea are clinical risk factors for fatal outcomes associated with coronavirus. These findings could be used for research, control, and prevention of the disease and could help providers take appropriate measures and improve clinical outcomes in these patients.

## Introduction

Coronavirus diseases 2019 (COVID-19) started in China in 2019 and then spread all over the world. The African continent has not been spared. As of July 21, 2022, the world counted 562 672 324 cases of COVID-19 with 9 176 657 deaths and Africa counted 8 711 651 cases with 178 575 deaths [1].

Limited access to healthcare and diagnostic capabilities, as well as restrictive testing policies, means that reported cases and deaths are estimated to be a fraction of the true numbers. A modeling study collating data from the WHO African region estimated that true COVID-19 deaths were three times higher than reported and only 1.4% of cases were actually reported [2]. Despite Africa being home to 17% of the global population, COVID-19-related deaths reported from Africa constitute only 4% of COVID-19-related deaths globally [3]. Several hypotheses exist for this comparatively lower COVID-19 mortality in Africa and it is unclear what proportion is due to underreporting versus host-factor differences, such as genetics, population age structures, and differences in comorbidities associated with mortality amongst an African population [4]. Although the majority of cases are mild, approximately 5% of patients are either severe or critical. Both critical and severe patients can progress to death if not properly managed [5].

Meta-analysis data have identified cardiovascular disease (CVD), advanced age, male sex, active smoking, chronic lung disease, diabetes, hypertension, obesity, cancer, acute renal failure, chronic kidney disease, acute inflammation, and organ damage as risk factors associated with COVID-19 mortality [6–9]. In the study by Lan Yang et al. among the common symptoms of COVID-19 infections, fatigue, sputum, hemoptysis, dyspnea, and chest tightness were independent predictors of death. Regarding laboratory examinations, a significant increase in pre-treatment absolute WBC count, LDH, PCT, D-Dimer, and ferritin, and a decrease in pre-treatment absolute lymphocyte count were associated with mortality [10].

While data on the risk factors for mortality amongst COVID-19 patients are available from other settings, there are limited data available from Africa. Mortality rates and factors associated with mortality amongst COVID-19 patients are poorly understood. Varying study designs and populations provide different estimates and impact sizes in African publications. Because of this unpredictability, a comprehensive and systematic review is needed. The objective of this study was to conduct a systematic review and meta-analysis to examine risk factors associated with mortality in COVID-19 patients in sub-Saharan Africa.

## Materials and methods

We followed the PRISMA (Preferred Reporting Items for Systematic Reviews and Meta-Analyses) guidelines (See S2 Checklist) and conducted a systematic review using PubMed, Google scholar, Europe PMC between January 1, 2020, and September 23, 2021.

### Eligibility criteria

The inclusion criteria for the studies were as follows: (1) subjects must be laboratory-confirmed COVID-19 patients; (2) studies must report risk factors or predictors of mortality in COVID-19 patients, with data available; (3) studies must be published in English; (4) studies must include multivariate analysis; and (5) studies must be conducted in the Sub-Saharan region. Exclusion criteria included case reports, review articles, commentaries, errata, protocols, abstracts, reports, letters to the editor, repeat studies, and studies without full text available. Clinical studies that did not clearly report death as an outcome were excluded. In addition, if two or more studies were published on the same patient sample by the same author, only the article with the highest quality was included.

### Information sources and search strategy

We conducted our literature search on Google Scholar, European PMC, Medline/PubMed as follows. We first searched our keywords individually (COVID-19, risk factors, mortality, sub-Saharan Africa*). For each keyword, we used synonyms: COVID-19 OR SARS-CoV-2; risk factors OR predictors; mortality OR lethality OR fatal outcome OR death; Sub-Saharan Africa* OR "Africa south of the Sahara "* OR Central Africa* OR West Africa* OR East Africa* OR Southern Africa* OR Benin* OR*Tanzania* OR Togo* OR Uganda* OR Zimbabwe* OR Cameroon* OR Cape Verde* OR Congo* OR Democratic Republic of Congo* OR Côte d'Ivoire* OR Ghana* OR Lesotho* OR Mauritania* OR Nigeria* OR Atlantic Islands* OR Senegal* OR Sudan* OR South Sudan*. Senegal* or Sudan* or South Sudan* or Swaziland* or Zambia* or Angola* or Botswana* or Gabon* or Mauritius* or Namibia* or Seychelles* or South Africa* or Equatorial Guinea* or Benin* or Burkina Faso* or Burundi* or Central African Republic* or Chad* or Comoros* or "Democratic Republic of Congo "* or Eritrea* or Ethiopia* or Gambia* or Guinea* or Guinea-Bissau* or Kenya* or Libya*. Bissau* OR Kenya* OR Liberia* OR Madagascar* OR Malawi* OR Mali* OR Mozambique* OR Niger* OR Rwanda* OR Sierra Leone* OR Somalia*. Next, we used the advanced search to combine the four keywords with AND between two keywords and OR between synonyms (strategy of search adapted from [11]). Two independent investigators reviewed the selected articles for abstract relevance. We evaluated the full text based on the inclusion and non-inclusion criteria.

### Study selection

Two investigators independently gathered the following information (BB and OO). They reviewed titles and abstracts to ensure that all publications included met the inclusion criteria (S1 Fig). The review rejected studies with missing, unclear, duplicated, or incomplete data.

### The data collection process and data items

The following information was extracted from each selected article by the investigators: first author, years of publication, country, number of patients, mean/median age, percentage of men, study design, percentage of COVID-19 patients with comorbidities or risk factors, effect estimates (hazard ratio (HR) or OR), and adjusted risk factors and vaccination status.

## Quality assessment

The methodological quality of the studies included in this meta-analysis was assessed using the methodological items for nonrandomized studies (MINORS) list. Studies were assessed by two independent reviewers (BB and OO). If a MIORS item was not reported, it received a score of 0; if it was reported but not adequate, it received a score of 1; and if it was reported and adequate, it received a score of 2. With 12 items, a maximum score of 24 points is available. If the overall MINORS score was 17 or more, we considered the study to be of high quality, and if the total score was $< 17$, we considered it to be of low quality [12, 13].

## Data synthesis and analysis

Key information such as study design and effect estimates were extracted directly from the original articles. We used published peer-reviewed ORs or HRs (and accompanying 95%CIs) to examine the association between the fatal outcome of COVID-19 and risk factors. Pooled HRs, or ORs, and 95% confidence intervals (CIs), were calculated separately to address mortality risk in patients with COVID-19. Heterogeneity was assessed by Cochran's Q statistic and the $I^2$ test. If no significant heterogeneity was observed ($I^2 \leq 50\%$, $P > 0.1$), a fixed-effects model was adopted; otherwise, a random effects model was applied [14]. Publication bias was assessed by the Egger test [15]. Subgroup analysis was performed to determine the source of heterogeneity. Data analyses were performed using Stata version 14. A p-value $< 0.05$ was considered significant.

# Results

## Study selection

In the PRISMA flowchart, we can see the approach used for the document search. Initially, we found 34263 articles. After eligibility criteria, 34251 were excluded. Of the excluded studies, 5010 articles were recorded after duplicates were removed, 4890 were perceived as unrelated searches, and 108 articles met the exclusion criteria. Finally, 12 published articles were included (Fig 1).

## Study characteristics and risk of bias

In the present systematic review and meta-analysis, the total number of patients was 43598. The characteristics of the included studies are described in Table 1. The studies were conducted in the DRC, Ethiopia, Guinea, Burkina Faso, Nigeria, South Africa, and Ghana. The majority of patients were admitted to hospitals, except in four studies where outpatients were also included. Most of the studies in the meta-analysis were retrospective cohorts. All studies were conducted in the period before December 2020. The number of confirmed COVID-19 cases in each study ranged from 25 to 10517. The majority of hospitalized patients were males. The proportion of patients with diabetes ranged from 3.1% to 36%, hypertension from 1.18% to 45.5%, heart disease from 2.22% to 24%, and chronic lung disease from 2.8% to 7%. As shown in Table 1, all articles included in the meta-analysis were of high quality according to the MINORS tool.

## COVID-19 mortality prevalence

Table 1 also shows the prevalence of patients who died from COVID-19. The prevalence of deaths was highest in DRC (29%) and lowest in Ethiopia (0.8%). Studies that included both inpatients and outpatients tended to have a low prevalence [18, 20, 22, 24]. The range of inpatient mortality varied from 4% to 29%, with the mean mortality of the patients who were

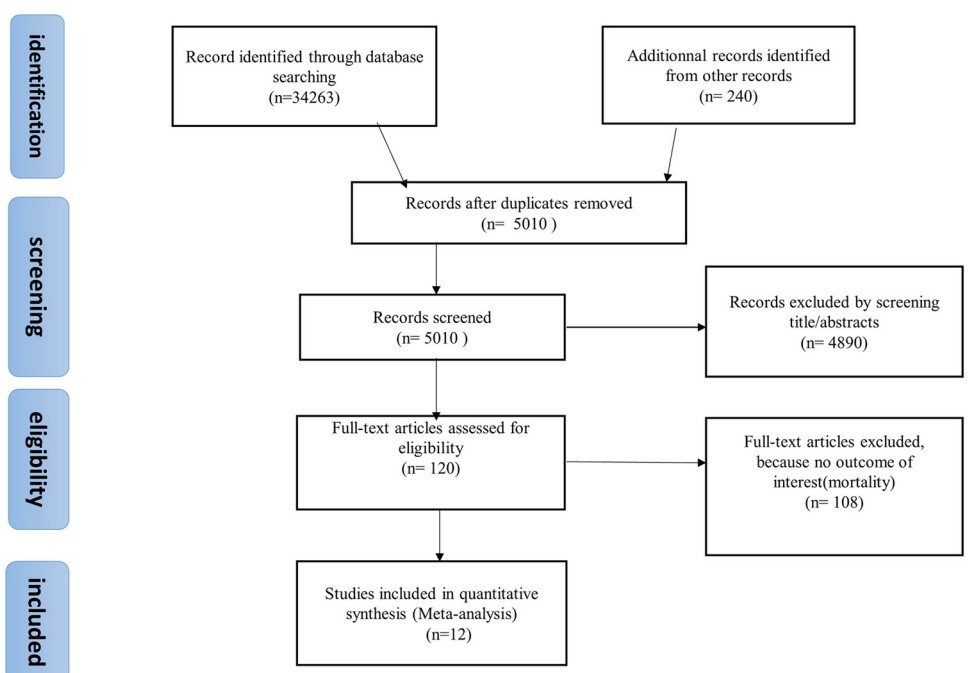

**Fig 1. Flowchart showing the selection of studies for the meta-analysis of the association of diabetes mellitus with COVID-19 mortality in sub-Saharan Africa.**

hospitalized of 7.3% (439/5972). The range of mortality for the combined studies (combining inpatients and community-diagnosed patients) varied from 0.8 to 9%, with a mean mortality rate of 4.4% (1664/37626). Separate data on the mean and range of mortality of patients treated or diagnosed in the community but not requiring admission were not clearly available in the included studies. The total number of patients who died was 2103. Thus, considering the total number of COVID-19 cases of 43598, the overall prevalence of deceased patients was 4.8%.

## Analysis of adjusted estimates

Adjusted data regarding mortality from COVID-19 infection was available in 11 studies [16–26] (Table 2). A multivariate Cox regression model was used in 3 studies, and a multivariate logistic regression model was applied in 8 studies. Predictors of mortality retained in multivariate analysis were advanced age [16, 17, 19, 22, 24–26], male sex [19, 20, 25], hypertension [19, 20, 22], diabetes mellitus [20, 22], obesity [17], chronic renal failure [17, 20, 22], and malignancy [18, 20]. Only one study reported HIV infection as an increased risk factor for death. Symptoms associated with an elevated risk of death were dyspnea [24–26], cough [24, 25], and vomiting [24]. One study, although unadjusted, showed elevated blood pressure as a predictor of mortality. Patients admitted at the severe stage also had an increased risk of death [19, 25, 26]. Among the included studies, the biological parameters associated with an increased risk of mortality in multivariate analysis were mainly elevated AST. None of the studies reported vaccination status (Table 2).

## Risk factors for mortality

Older age was one of the risk factors for death, and the pooled HR and OR for coronavirus mortality were 9.01 (95% CI: 6.30–11.71) and 1.04 (95% CI: 1.02–1.06), respectively (Table 3

**Table 1. Characteristics of the included study.**

| | | | | | | | | | | | | | |
|---|---|---|---|---|---|---|---|---|---|---|---|---|---|
| **Studies including only hospitalized patients** | | | | | | | | | | | | | |
| Authors | Design | Country | Time period of study | size | Male (%) | Age [median, (IQR), mean (SD)] | Mortality rate (%) | Number of deaths | DM (%) | HTN (%) | CVD (%) | Chronic lung diseases (%) | MINORS |
| Matangila et al. [16] | retrospective cohort | DRC | March 11-July 22, 2020 | 160 | 51.0 | 54 (38–64) | 20 | 32 | 19 | 34 | 7 | 3 | 17 |
| Nachega et al. [17] | retrospective cohort | DRC | March 10-July 31, 2020 | 766 | 65.6 | 46 (34–58) | 13.2 | 101 | 14 | 25.4 | 3.9 | 3.4 | 18 |
| Jaspard et al. [19] | Prospective cohort | Burkina and Guinea | April 1-november 4 (Guinea) March 1-November 12, 2020(Burkina) | 1805 | 64.0 | 41 (30–57) | 5 | 90 | 12 | 21 | - | - | 18 |
| Laura Skrip et al.[21] | retrospective cohort | Burkina | Through 10 May 2020 | 751 | 57.1 | | 6.5 | 49 | 212 | 45.5 | - | - | 18 |
| Donamou et al. [23] | retrospective cohort | Guinea | March 12-July 12, 2020 | 140 | 79 | 58±14 | 25 | 35 | - | - | - | - | 17 |
| Abayomi et al. [25] | retrospective cohort | Nigeria | Submit in september 2, 2020 | 2184 | 65.8 | 43±16 | 4 | 87 | 6.8 | 16.7 | - | - | 18 |
| Bepouka et al. [26] | retrospective cohort | DRC | March 23-June 15, 2020 | 141 | 67.4 | 49.6±16.5 | 29 | 41 | 17 | 23.4 | 4.6 | - | 17 |
| Authors | Design | Country | Time period of study | Size | Male (%) | Age [median, (IQR), mean (SD)] | Mortality rate (%) | Number of deaths | DM (%) | HTN (%) | CVD (%) | CLD (%) | MINORS |
| Boateng et al. [27] | retrospective cohort | Ghana | June 1st-July 27 th, 2020 | 25 | 56 | 59.3±20.6 | 16 | 4 | 36 | 72 | 24 | - | 17 |
| Totals | NA | NA | NA | 5972 | NA | NA | NA | 439 | NA | NA | NA | NA | NA |
| **Studies including hospitalized and community patients** | | | | | | | | | | | | | |
| Authors | Design | Country | Time period of study | Size | Male (%) | Age [median, (IQR), mean (SD)] | Mortality rate (%) | Number of deaths | DM (%) | HTN (%) | CVD (%) | CLD (%) | MINORS |
| Abraha et al.[18] | Retrospective cohort | Ethiopia | May 10-october 16, 2020 | 2617 | 63.3 | 29(24–38) | 0.8 | 21 | 3.1 | 3.1 | - | 2.8 | 18 |
| Osibogun et al.[20] | Retrospective cohort | Nigeria | February 27-July 6, 2020 | 2184 | 65.8 | 43(35–55) | 3.3 | 72 | 6 | 1.18 | 2.22 | - | 18 |
| Boule et al. [22] | Retrospective cohort | South Africa | March 1- June 9, 2020 | 22308 | 46 | - | 2.8 | 625 | 11 | 236 | - | 7 | 18 |
| Elimian et al.[24] | Retrospective cohort | Nigeria | February 27June 8, 2020 | 10517 | 67.7 | 35.6±15 | 9 | 946 | - | - | - | - | 17 |
| **Totals** | NA | NA | NA | 37626 | NA | NA | NA | 1664 | NA | NA | NA | NA | NA |

Abbreviations: SD: standard deviation; IQR: interquartile range; DM: diabetes mellitus; HTN: hypertension; CVD: cardiovascular diseases; CLD: chronic lung diseases; MINORS: the methodological items for nonrandomized studies; DRC: the Democratic Republic of the Congo; NA: not applicable

and Fig 2). Male patients had a significantly increased risk of COVID-19-related mortality, according to four studies (pOR 1.52; 95 percent CI 1.04–2) (Table 2 and Fig 3). Patients with chronic renal failure (pHR 1.87; 95% CI 1.45–2.29), hypertension (pOR 2.14; 95% CI 1.28–3.01), severe and critical COVID-19 stage at admission (pOR 9.04; 95% CI 3.14–14.94), cough (pOR 2; 95% CI 1.34–2.66), and dyspnea all had a significantly increased risk of coronavirus-related mortality (pOR 6.25; 95% CI 3.90–8.61) (Figs 4–9). But in the meta-analysis, diabetes mellitus was not associated with mortality (pHR 1.14; 95% CI 0.63–1.66; pOR 1.51; 95% CI 0.79–2.22) (Table 3).

**Table 2. Risk factors for increased risk of mortality in studies using regression models.**

| Authors | Setting (Source of the cohort of case) | Regression model | Significant risk factors (effect estimate, 95% CI) |
|---------|----------------------------------------|------------------|---------------------------------------------------|
| Nachega et al. | 7 largest health facility in Kinshasa (hospital admitted) | Cox regression | age < 20 years (adjusted hazard ratio [aHR] = 6.62, 95% CI: 1.85–23.64), 40–59 years (aHR = 4.45, 95% CI: 1.83–10.79), and [3] 60 years (aHR = 13.63, 95% CI: 5.70–32.60) compared with those aged 20–39 years, with obesity (aHR = 2.30, 95% CI: 1.24–4.27), and with chronic kidney disease (aHR = 5.33, 95% CI: 1.85–15.35) |
| Abraha et al. | Covid-19 isolation and treatment center, in Mekelle city (community and hospital admitted) | multi-variate regression analysis [adjusted relative risk (aRR)] was undertaken (with backward stepwise elimination), | older age (aRR 2.37, 95% CI 1.90–2.95; P < 0.001), malignancy (aRR 6.73, 95% CI 1.50–30.16; P = 0.013) and surgery/trauma (aRR 59.52, 95% CI 12.90–274.68; P < 0.0001). |
| Jaspard et al. | 3 hospitals in Burkina and Guinea (hospital admitted) | multivariable logistic regression | In multivariable analysis, the risk of death was higher in men (aOR 2.0, 95% CI 1.1; 3.6), people aged 60 years (aOR 2.9, 95% CI 1.7; 4.8) and those with chronic hypertension (aOR 2.1, 95% CI 1.2; 3.4). |
| Osibogun et al. | 10 isolation and treatment facilities in Lagos (community and hospital admitted) | multivariable logistic regression model | hypertension (OR: 2.21, 95%CI: 1.22–4.01), diabetes (OR: 3.69, 95% CI: 1.99–6.85), renal disease (OR: 12.53, 95%CI: 1.97–79.56), cancer (OR: 14.12, 95% CI: 2.03–98.19) and HIV (OR: 1.77–84.15] |
| Laura Skrip et al. | Health center, in Ouagadougou, Burkina Faso (hospital admitted) | logistic regression. | the odds of mortality for cases not receiving oxygen therapy were significantly higher than for those receiving oxygen, such as due to disruptions to standard care (OR 2.07; 95% CI 1.56–2.75). Cases receiving convalescent plasma had 50% reduced odds of mortality than those who did not (95% CI 0.24–0.93 |
| Matangila et al. | Single center in Kinshasa (hospital admitted) | Multivariate logistic regression models | OR: Older age: 1.06(1.0–1.11), lower SpO2: 0.94(0.90–0.98), higher heart rate: 1.06(1.02–1.11), elevated AST:1.02(1.01–1.03) |
| Bepouka et al. | Single center in Kinshasa (hospital admitted) | COX regression models. | age between 40 and 59 years [adjusted Hazard Ratio (aHR) (aHR): 4.07; 95% CI: 1.16–8.30], age at least 60 years (aHR: 6.65; 95% CI: 1.48–8.88), severe or critical COVID-19 (aHR: 14.05; 95% CI: 6.3–15.67) and presence of dyspnea (aHR: 5.67; 95% CI: 1.46–21.98) |
| Boule et al. | electronic clinical information systems used in all public sector health facilities in the Western Cape (community and hospital admitted) | Cox-proportional hazards models adjusted for age, sex, location and comorbidities | male sex, increasing age, diabetes, hypertension and chronic kidney disease |
| Boateng et al. | treatment centre of the University Hospital, Kumasi, Ghana (hospital admitted) | multivariate logistic regression modelling | Increasing age and high systolic blood pressure in unadjusted but no factors in multivariate analysis |
| Danamou et al. | Intensive Care Unit of the COVID Treatment Center of Donka National Hospital, (hospital admitted) | multivariate logistic regression analysis | Acute Respiratory Distress Syndrome (ARDS) (OR = 6.33, 95% CI [1.66–29]; p = 0.007), a Brescia score ≥ 2 (OR = 5.8, 95% CI [1.7–19.2]; p = 0.004) and admission delay (OR = 5.6, 95% CI [1.8–17.5]; p = 0.003). |
| Abayomi et al. | nine treatment centres in Lagos state, Southwest Nigeria (hospital admitted) | multivariable logistic regression models | Difficulty in breathing was the most significant symptom predictor of COVID-19 death (OR:19.26 95% CI 10.95–33.88). |
| Elimian et al. | Nigeria surveillance and laboratory data (community and hospital admitted) | Multivariable logistic regression analysis | aged ≥51 years, patients in farming occupation (aOR 7.56, 95% CI 1.70 to 33.53) and those presenting with cough (aOR 2.06, 95% CI 1.41 to 3.01), breathing difficulties (aOR 5.68, 95% CI 3.77 to 8.58) and vomiting (aOR 2.54, 95% CI 1.33 to 4.84). |

Abbreviations: aHR: adjusted hazard ratio aOR: adjusted odds ratio aRR: adjusted risk ratio OR: odds ratio HIV: human immunodeficiency virus CI: confidence interval

sp O2: oxygen saturation AST: aspartate aminotransferase ARDS Acute respiratory distress syndrom

Note: None of the studies reported vaccination status

**Table 3. Results of subgroup analysis based on demographic, clinical, and comorbidities variables associated with coronavirus mortality.**

| Risk factors | Effect mesures | Numbers of study | Effect size (95%) | Heterogeneity $I^2_{value}$ | Egger's test | P |
|---|---|---|---|---|---|---|
| **Advanced age** | pHR | 3 | 9.01(6.30–11.71) | 41.9 | 0.17 | 0.157 |
| | p OR | 6 | 1.04(1.02–1.06) | 82.7 | 0.000 | 0.060 |
| **Male** | | | | | | |
| | pOR | 4 | 1.52(1.04–2) | 0 | 0.517 | |
| **CKD** | pHR | 2 | 1.87(1.45–2.29) | 1.1 | 0.315 | NA |
| | p OR | 1 | 12.53(1.97–79.56) | | - | - |
| **HTN** | pHR | 2 | 1.02(0.59–1.45) | 0 | 0.891 | NA |
| | p OR | 2 | 2.14(1.28–3.01) | 0 | 0.903 | NA |
| **DM** | pHR | 2 | 1.14(0.63–1.66) | 0 | 0.748 | NA |
| | p OR | 2 | 1.51(0.79–2.22) | 70.5 | 0.065 | NA |
| **Severe or critical COVID-19** | pHR | 1 | 14.05(6–15.67) | - | - | - |
| | p OR | 2 | 9.04(3.14–14.94) | 0 | 0.442 | NA |
| **Cough** | p OR | 2 | 2(1.34–2.66) | 0 | 0.792 | NA |
| **Dyspnea** | pHR | 1 | 5(1.46–21.98) | - | - | - |
| | p OR | 2 | 6.25(3.90–8.61) | 80.6 | 0.023 | NA |

Abbreviations: p HR: pooled hazard ratio; p OR: pooled odds ratio; NA: not available; HTN: hypertension; CKD: chronic kidney disease; DM: diabetes mellitus

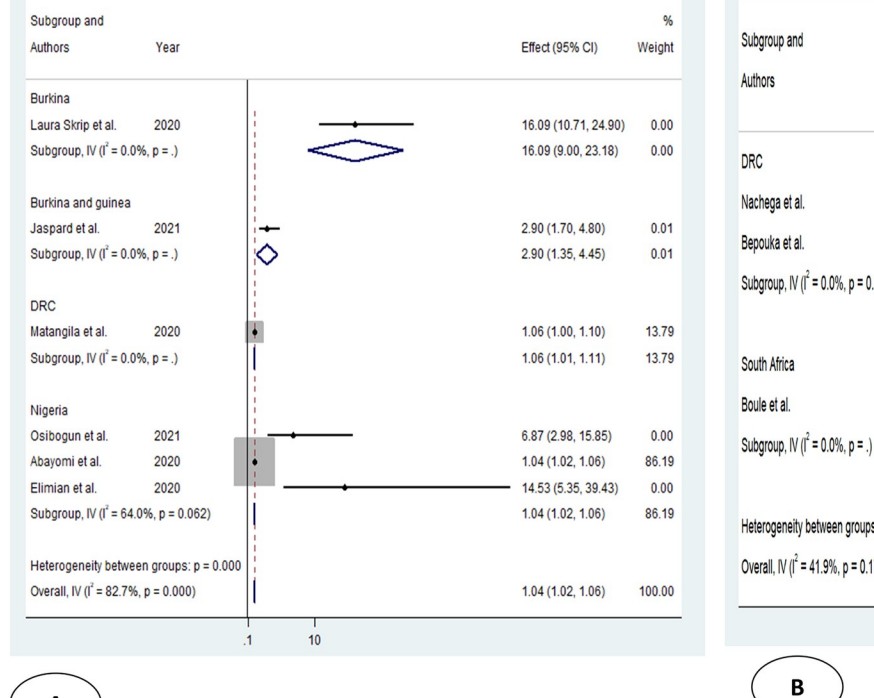 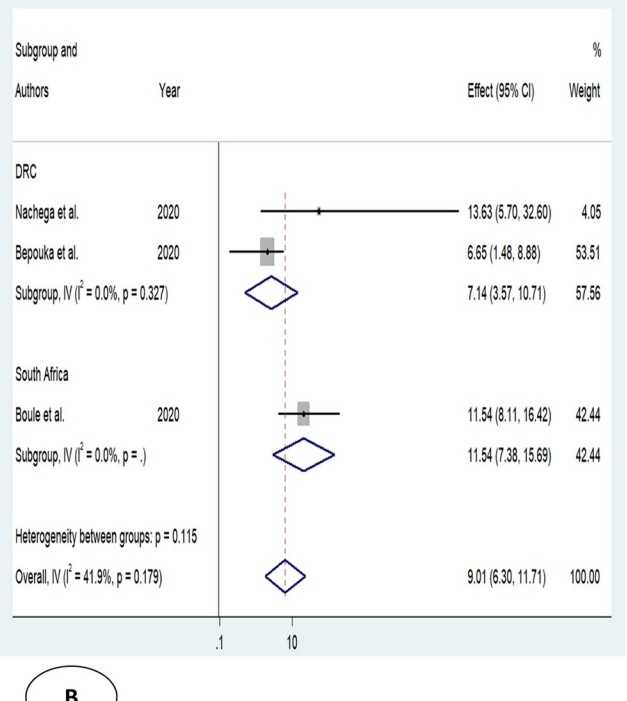

**Fig 2. Forest plot showing the estimate for advanced age on COVID-19 mortality.** Forest plots of studies using A. Odds ratio, B. Hazard ratio.

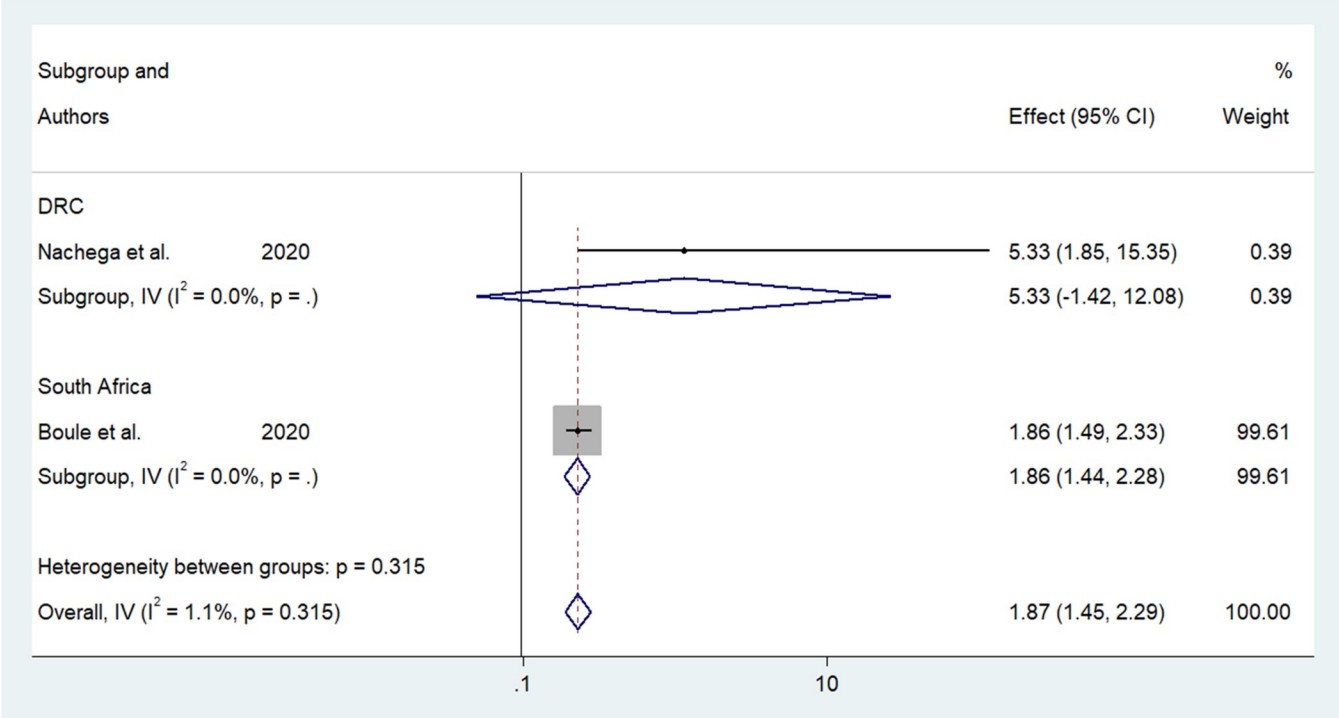

**Fig 3. Forest plot showing the estimate for the effects of CKD on COVID-19 mortality.**

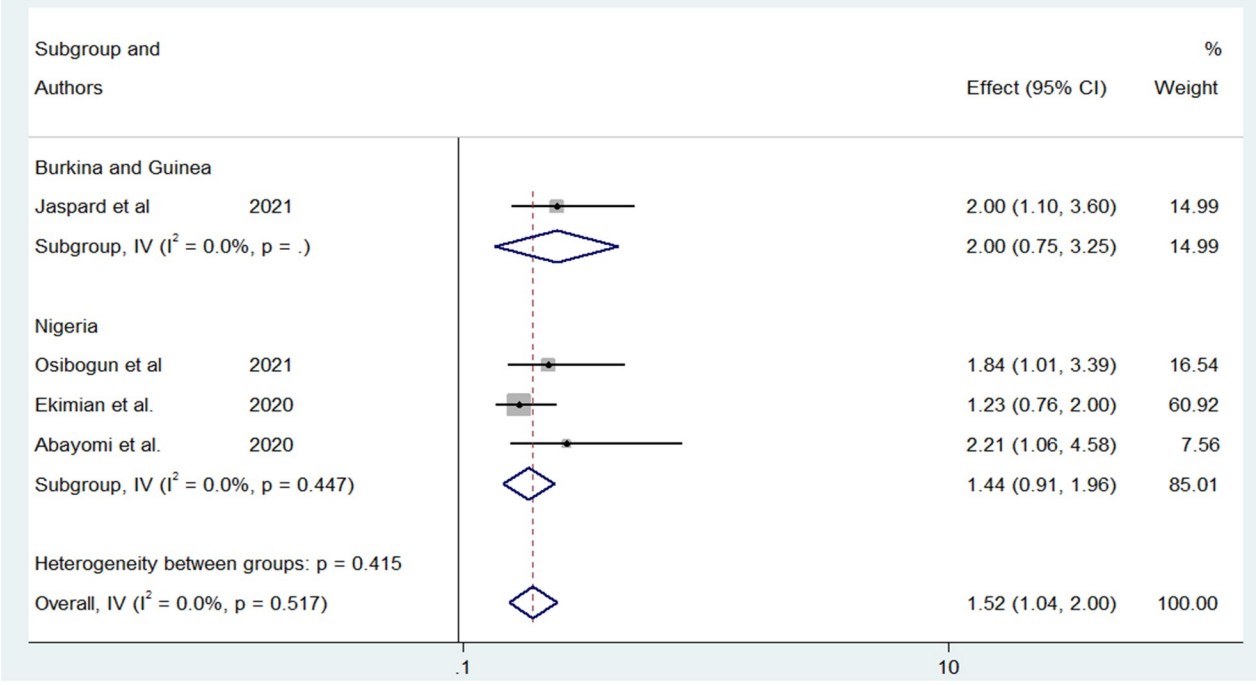

**Fig 4. Forest plot showing the estimate for the effects of sex on COVID-19 mortality.**

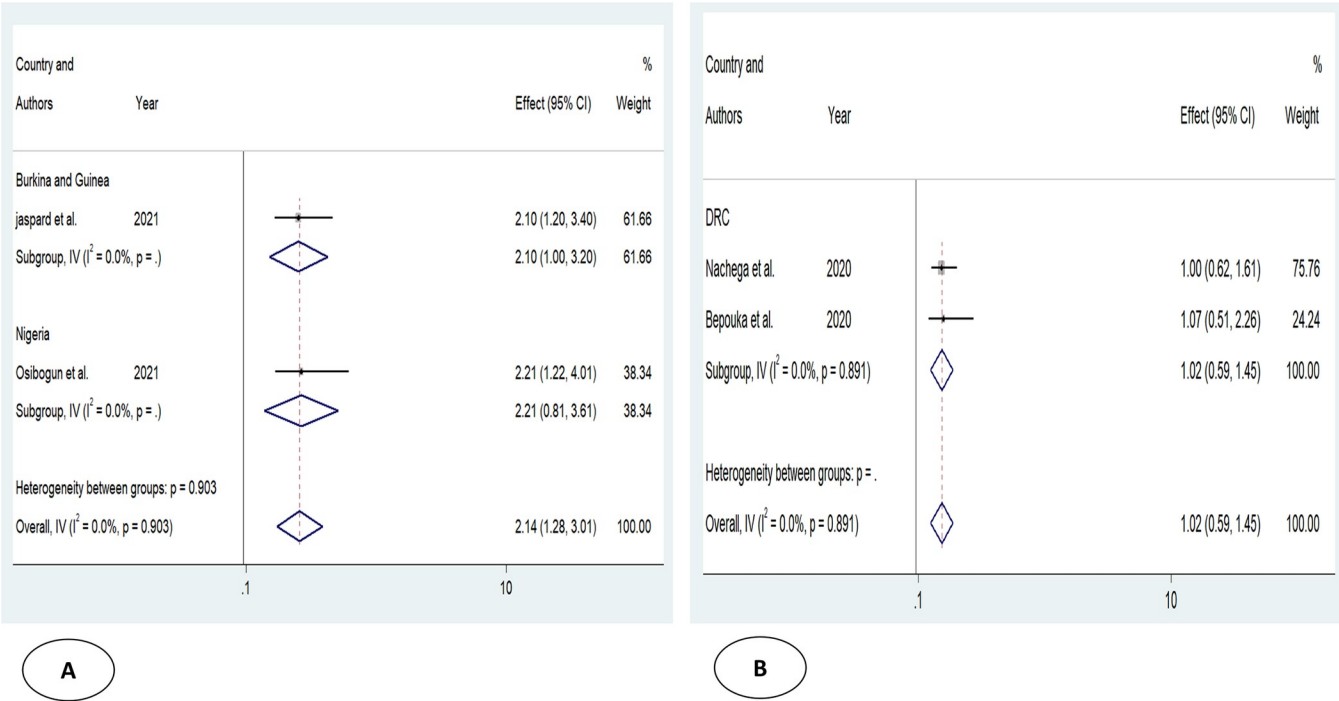

**Fig 5. Forest plot showing the estimate for the effects of hypertension on COVID-19 mortality.** Forest plots of studies using A. Odds ratio, B. Hazard ratio.

## Publication bias and heterogeneity

Pool OR statistics for age, diabetes mellitus, and dyspnea showed heterogeneity between the studies considered ($I^2$ with values of 82.7, 70.5, and 80.6, respectively. Subgroup analysis was not performed for diabetes mellitus and dyspnea as there were only two studies included. With

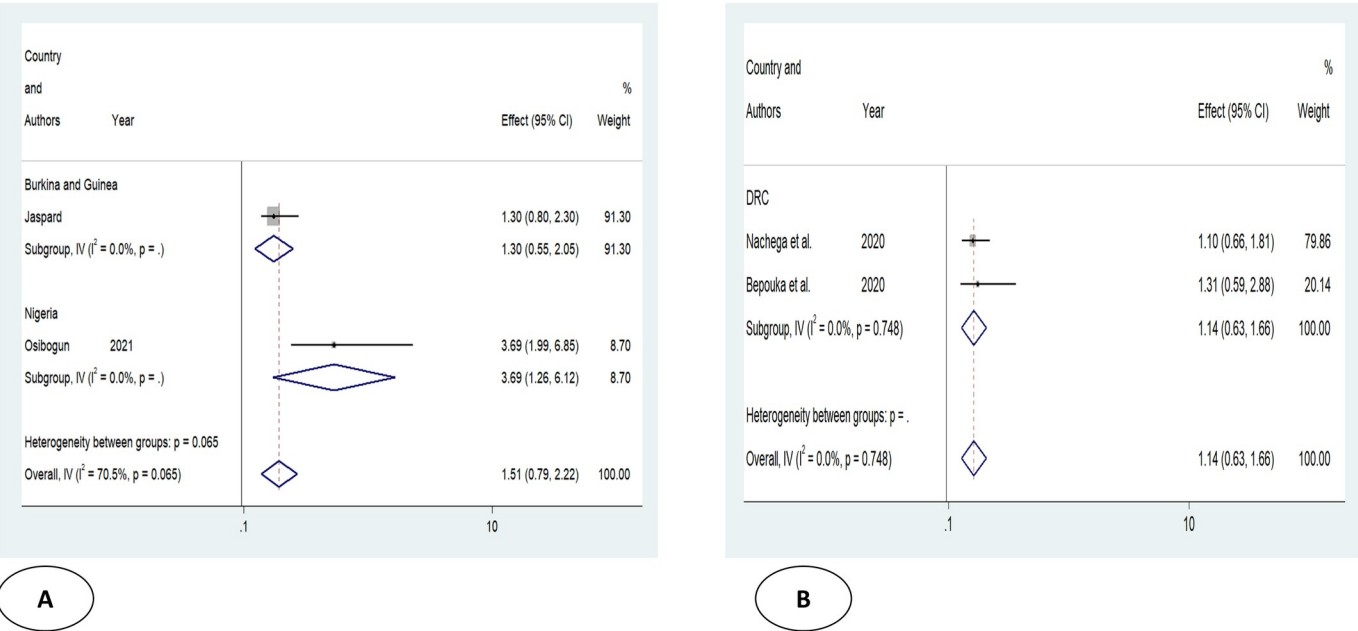

**Fig 6. Forest plot showing the estimate for the effects of DM on COVID-19 mortality.** Forest plots of studies using A. Odds ratio, B. Hazard ratio.

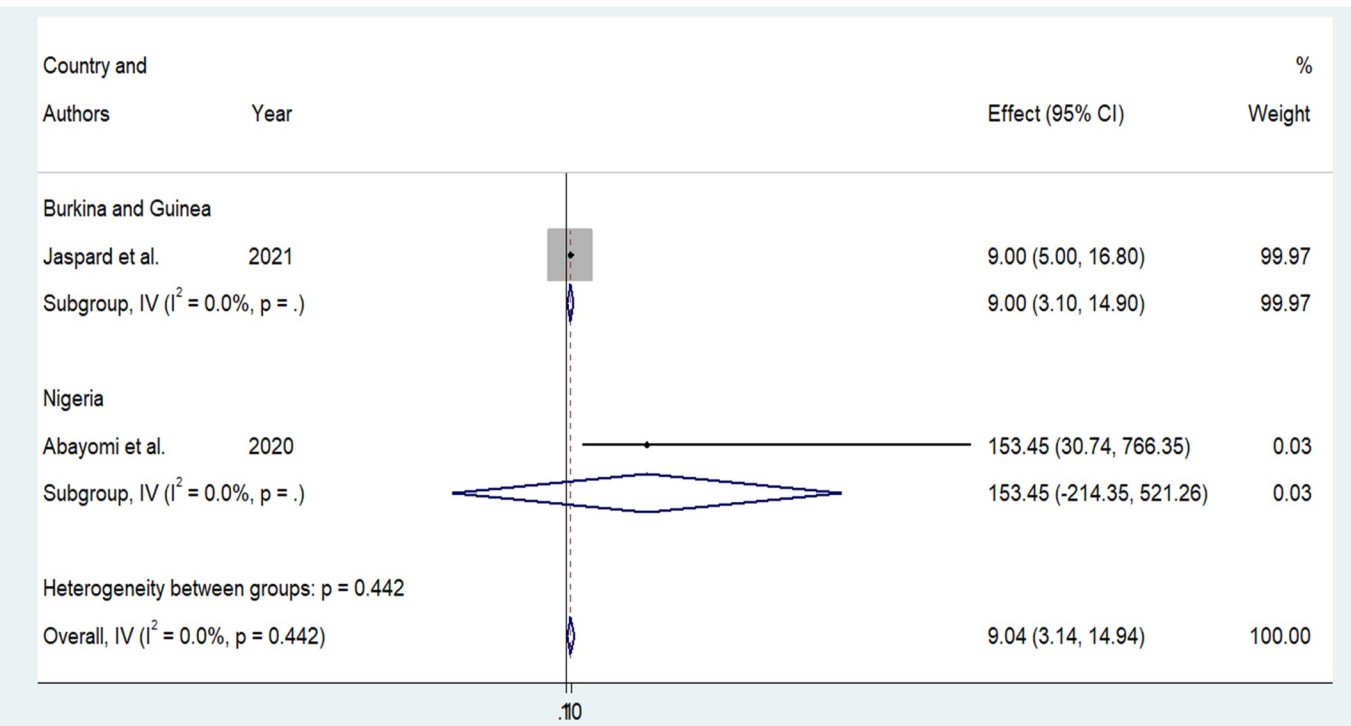

**Fig 7. Forest plot showing the estimate for the effects of severe or critical on COVID-19 mortality.**

respect to age, by performing subgroup analysis, dividing the studies into two groups (studies where age ranges were well specified, 60 or more and less than 60 years; studies where age ranges were not well specified, either using median or mean age). By performing this subgroup analysis, studies where age ranges were subdivided into 60 or older and younger than 60 years no longer showed heterogeneity. (Figs 10 and 11).

## Discussion

The mortality prevalence of covid-19 patients in SSA was 4.8%. Advanced age, male gender, chronic kidney diseases, hypertension, severe or critical condition on admission, coughing,

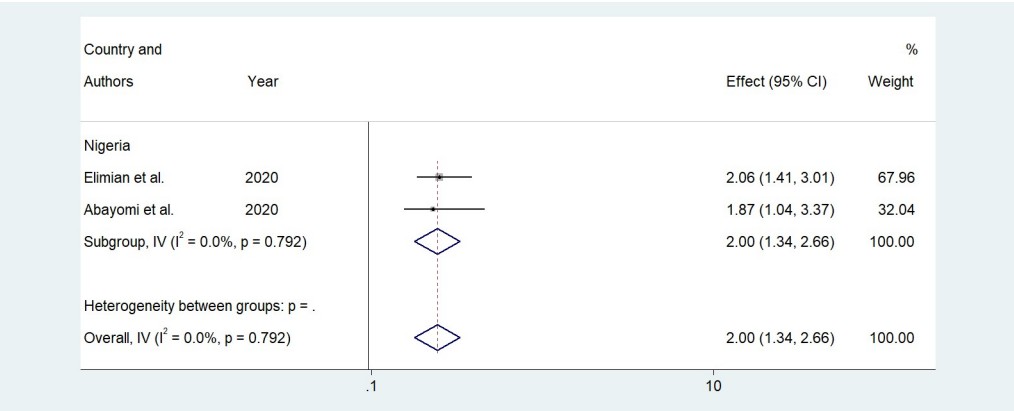

**Fig 8. Forest plot showing the estimate for the effects of cough on COVID-19 mortality.**

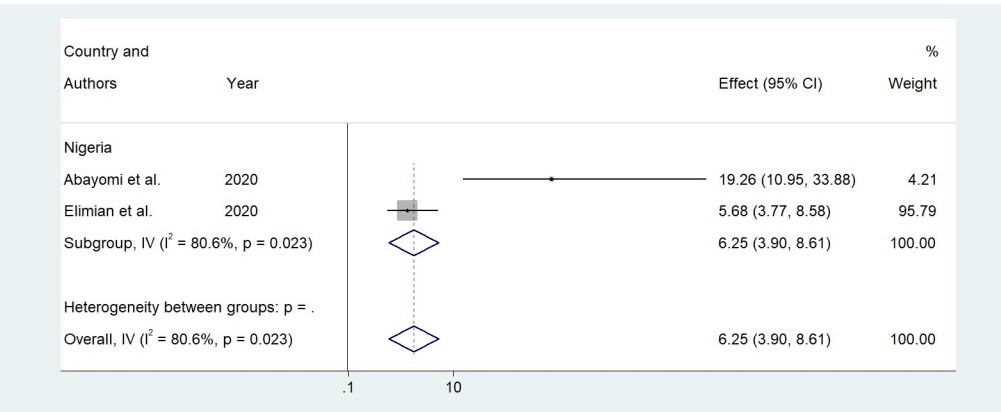

**Fig 9. Forest plot showing the estimate for the effects of dyspnea on COVID-19 mortality.**

and dyspnea were clinical risk factors associated with a fatal outcome in COVID 19. The mortality prevalence of 4.8% in our study is higher than the mortality reported by the WHO. The reason for this is that the majority of the included patients in our meta-analysis were hospitalized, and in some included studies, the proportion of severe and critical patients was very high. The emergence of variants of concern may also influence case numbers and mortality, although almost all of the included studies were conducted in the first wave when these variants were not yet formally reported. Patient mortality in the DRC ranged from 13–29% versus 1.5% according to WHO data. In Ghana, mortality was 16% versus 0.87% according to WHO data. In Nigeria, mortality ranged from 3.3 to 9% versus 1.2% according to WHO data. In these countries, our meta-analysis shows high prevalence because most of the included patients are hospitalized, with a high proportion of severe cases. Mortality in South Africa and Ethiopia was close to WHO data (0.8 and 2.8 versus 2.5 and 1.5) because the studies included in these two countries also included cases from the community [1]. In addition, considering overall mortality was 4.8%; The range of inpatient mortality was 4–29% with a mean inpatient mortality of 7.3% (439/5972). Considering the range of mortality being 4–29%, one would expect higher mortality than 4.8%. This value of 4.8% is due to the fact that it is overall mortality combining inpatients and community patients, this value of 4.8% is due to the fact that it is the overall mortality combining inpatients and community patients, whose number is not negligible.

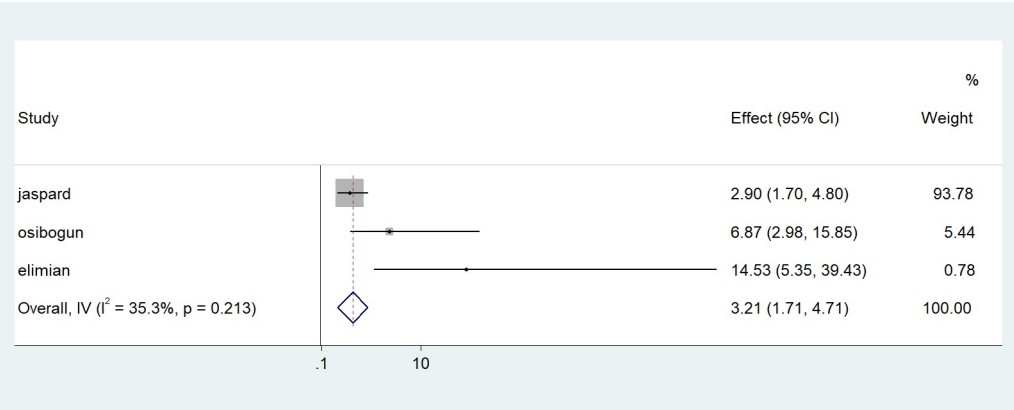

**Fig 10. Forest plot showing the estimate for the effects of age (≥60 vs <60) on COVID-19 mortality.**

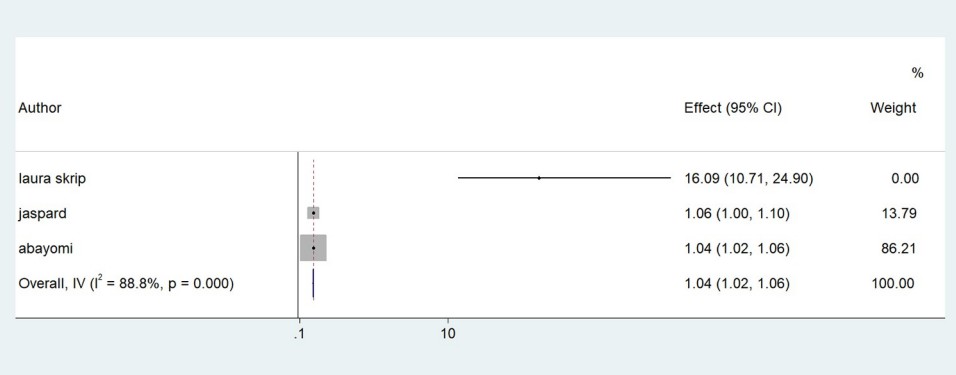

**Fig 11. Forest plot showing the estimate for the effects of high median or mean or no on COVID-19 mortality.**

Advanced age has previously been reported as a factor associated with mortality in COVID 19 [6, 7, 9, 28, 29] but not as a mortality risk factor by two independent meta-analyses [8, 10]. One plausible reason could be that aging affects the functions of immune cells: T cells (CD4+, CD8+), and B cells [7, 30]. T-cell and B-cell clonal diversity are reduced at advanced age, which has been associated with decreased immune responses to viral infections such as influenza [7, 31]. In addition, a significant increase in type 2 cytokines could result in lasting proinflammatory immune responses with subsequent poor disease outcome [7, 32].

Male gender was retained as a mortality risk factor for COVID-19 patients. Male gender has been controversial with most previous meta-analyses in support [6, 7, 9, 29] and not Mehraeena E et al. [8]. Two plausible explanations have emerged from the literature. First, in the adaptive immune system, males have a lower T-cell (CD8+, CD4+) and B-cell count as compared to females [7, 33, 34]. Second, females have the advantage of the immune regulatory genes located on X chromosome with a higher expression of Toll-like receptor-7 (TLR7) [7, 35] and a differential expression of carcinoembryonic antigen 2 (CEA2) [36, 37].

CKD was retained as a risk factor for the mortality of COVID-19 patients (Table 3). Two previous meta-analyses have echoed our finding [28, 29] while the meta-analysis of Xu J and Mehraeena E et al. have not [6, 8]. Patients with CKD, have increased levels of pro-inflammatory cytokines. The resulting increased oxidative stress drives an inflammatory immune response. An impaired immune system may increase susceptibility to bacterial and viral pulmonary infections [38, 39].

HTA was identified as a risk factor for mortality in COVID-19 patients. This result was already confirmed by previous meta-analyses [7, 9, 28] but not retained by the meta-analysis of Xu J, Mehraeena E et al. [6, 8] The mechanism underlying the association between preexisting hypertension and COVID-19 are to be fully elucidated. Nevertheless, endothelial dysfunction and Renin–Angiotensin System (RAS) imbalance have both been incriminated in severe outcome of COVID-19 in hypertension. Hypertension has been associated with endothelial dysfunction and a proinflammatory state i.e. higher Angiotensin II levels, chemokines, and cytokines, including interleukin-6 (IL-6) and tumor necrosis factor-α (TNF-α) [40, 41]. COVID 19 outcome is poor when conventional RAS axis (ACE/Ang II/AT1R) is activated and unconventional axis (ACE2/Ang 1-7/Mas) is downregulated [40, 42]. RAS imbalance also promotes a proinflammatory state, which is a suggested key pathophysiological mechanisms of COVID-19 [43, 44].

Either a severe or critical stage of COVID 19 at consultation or hospital admission was retained as a risk factor for mortality in COVID-19 patients. Previous meta-analysis has found

a similar result [8]. A severe or critical stage at hospital admission reflects late presentation, which is the single most important predictor of poor outcomes in COVID 19. Effective home management of COVID-19 may therefore play a critical role in reducing late presentation and death. McCullough et al. have suggested, that public awareness of the effects of late presentation be raised consistently through social mobilization and communication for behavior change. Many studies in our meta-analysis were conducted in COVID 19 treatment centers and were therefore not done at the early stages of the disease. Even in developed countries, patients waited until they were very ill to be tested for COVID 19 at the hospital [25, 45].

Cough was retained as a predictor of mortality in COVID-19 patients as confirmed in two previous meta-analyses [8, 10]. Similar to other viral infections, patients with COVID-19 may present with fever, cough, muscle pain, fatigue, headache, gastrointestinal symptoms, and dyspnea [46]. Cough is important in triage of suspected COVID-19 for the referral diagnosis. Coughing and/or dyspnea were both identified as mortality risk factors in COVID-19 patients. This result was already confirmed in previous meta-analyses [10, 29]. Dyspnea was correlated with higher mortality, even after adjusting for age, gender, and other confounding factors [47, 48]. Dyspnea is an important clinical factor to gauge the high risk of fatal outcome in COVID 19. Blood gas analysis or saturation is a useful tool in determining the severity of dyspnea. A low $SpO_2$ may reflect a more severe dyspnea with an increased mortality risk of COVID 19 [29]. Even if happy hypoxia i.e., low $SpO_2$ without dyspnea has been associated with poor vital outcomes in COVD 19 [49].

None of the studies reported vaccination status. The included studies were conducted in the first wave when vaccination was not yet effective. Vaccination could help reduce this high inpatient mortality, especially if vaccination coverage was high. In the literature, there is evidence of the efficacy of COVID-19 vaccines against severe forms and a reduction in mortality. Although most doses of COVID-19 vaccines have been administered in high and middle-income countries. The efficacy of the vaccines has been demonstrated in several countries where, for example, the efficacy of the two doses was 92%, 96%, and 89%, respectively, in Israel, the USA, and the UK [50]. But in Africa, up to the end of 2021, vaccination coverage was only 17% [51].

## Limitations

Although this systematic review presented pooled estimates from 12 studies across SSA, our study has some limitations. First, the sample size of some of the included studies was small, which is one of the possible factors influencing COVID-19 mortality. Second, statistically significant results were more likely to be accepted and published in similar studies than non-statistically significant results. But in fact, the data in the meta-analysis came mainly from studies that were published, which may lead to publication bias. Third, the causal relationship between risk factors and poor outcome in patients with COVID-19 cannot be confirmed because of the inherent limitations of the observational study. Therefore, well-designed studies with larger sample sizes are needed for verification.

## Implications of the results for practice, policy and future research

From a research perspective, making these data available could help advance infectious disease research and stimulate the development of reliable data-based risk prediction algorithms. Our findings can help researchers and policymakers modify management options for COVID-19 patients.

From a patient care perspective, evidence-based risk stratification in patient care could help plan resources and discover trends that predict which patients require special attention and

should be managed promptly. Individuals with risk factors, such as patient features, will be swiftly recognized based on the findings of this study. Similarly, in individuals with advanced disease, the choice of certain more aggressive treatments could be influenced by the prediction of mortality risk.

From a public health perspective, our findings show the importance of preventing late presentation (especially in those with underlying risk factors) for target health promotion messaging.

## Conclusion

This meta-analysis provides evidence of correlations between important prognostic factors and survival in patients with COVID-19. Clinicians and other healthcare providers should consider these factors when discussing the expected prognosis of patients with COVID-19 and take appropriate action accordingly. Further studies are needed to better understand the pathophysiologic mechanisms of the association between these predictors and COVID-19 infection.

## Supporting information

**S1 Checklist. Prisma for abstract checklist.**
(DOCX)

**S2 Checklist. Prisma checklist.**
(DOCX)

**S1 Fig. Prisma flow diagram.**
(DOCX)

## Acknowledgments

The authors would like to express our gratitude to the editor and reviewers for their insightful criticism and suggestions, which raised the quality of the article. The authors acknowledge Prof Modibo Sangare and Siobhan Johnstone for editing the first version manuscript.

## Author Contributions

**Conceptualization:** Ben Bepouka.

**Data curation:** Ben Bepouka, Hippolyte Situakibanza.

**Formal analysis:** Ben Bepouka.

**Investigation:** Ben Bepouka, Ossam Odio.

**Methodology:** Ben Bepouka.

**Project administration:** Ben Bepouka, Hippolyte Situakibanza.

**Resources:** Ben Bepouka, Hippolyte Situakibanza.

**Software:** Ben Bepouka.

**Supervision:** Marcel Mbula, Hippolyte Situakibanza.

**Validation:** Ben Bepouka.

**Visualization:** Ben Bepouka, Ossam Odio.

**Writing – original draft:** Ben Bepouka, Jean Marie Kayembe.

**Writing – review & editing:** Ben Bepouka, Nadine Mayasi, Madone Mandina, Murielle Long-okolo, Ossam Odio, Donat Mangala, Marcel Mbula, Jean Marie Kayembe, Hippolyte Situakibanza.

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
