## [Decision Letter · Decision Letter 0]

23 Jun 2022

PONE-D-21-36467

Risk factors for in-hospital mortality in COVID-19 patients in sub-Saharan Africa: A systematic review and meta-analysis

PLOS ONE

Dear Dr. Bepouka,

Thank you for submitting your manuscript to PLOS ONE. After careful consideration, we feel that it has merit but does not fully meet PLOS ONE’s publication criteria as it currently stands. Therefore, we invite you to submit a revised version of the manuscript that addresses the points raised during the review process.

Specifically, the reviewer mentioned a couple of concerns in Results section including a likely miscalculation in the mortality numbers. Please have all the comments addressed point-by-point.

We look forward to receiving your revised manuscript.

Kind regards,

Jianhong Zhou

Staff Editor

PLOS ONE

2. Please include a caption for figures 5 to 11.

4. Thank you for submitting the above manuscript to PLOS ONE. During our internal evaluation of the manuscript, we found significant text overlap between your submission and the following previously published works, some of which you are an author.

- https://pubmed.ncbi.nlm.nih.gov/34238232/

- https://bmcinfectdis.biomedcentral.com/articles/10.1186/s12879-021-06369-0

- https://pubmed.ncbi.nlm.nih.gov/33075534/

- https://www.frontiersin.org/articles/10.3389/fphys.2021.665064/full

- https://link.springer.com/article/10.1007/s11255-020-02740-3?code=aa12583d-3699-4ff6-8f1f-78915123c553&error=cookies_not_supported

- https://bmcinfectdis.biomedcentral.com/articles/10.1186/s12879-021-06536-3

Please revise the manuscript to rephrase the duplicated text, cite your sources, and provide details as to how the current manuscript advances on previous work. Please note that further consideration is dependent on the submission of a manuscript that addresses these concerns about the overlap in text with published work.

We will carefully review your manuscript upon resubmission, so please ensure that your revision is thorough

Reviewers' comments:

Reviewer's Responses to Questions

**Comments to the Author**

1. Is the manuscript technically sound, and do the data support the conclusions?

Reviewer #1: Partly

2. Has the statistical analysis been performed appropriately and rigorously? 

Reviewer #1: Yes

3. Have the authors made all data underlying the findings in their manuscript fully available?

Reviewer #1: Yes

4. Is the manuscript presented in an intelligible fashion and written in standard English?

Reviewer #1: Yes

5. Review Comments to the Author

Reviewer #1: Methodology: The paper is well written and the methodology of the search strategy sound.

The paper is relevant and timely and provides information from subSaharan Africa that is sadly lacking

Results: The mortality of different COVID-19 variants is likely to be different. Also the mortality of infected individuals is likely to be affected by vaccination rates. I believe Table 1 needs to have the time period of study, and perhaps a comment (in table 2 as to the vaccination status during the period of study.

Next I am surprised by your mortality rate of 3%. I agree table 1 shows 43598 cases of Covid-19 from the relevant countries but I calculated from your percentage mortality figures and got 2555 deaths which is 5.8%. I think to clarify you need a column that includes the number of deaths as well as the mortality percent.

Table 1 should probably include also the source of the cohort of cases - eg community, hospital admitted and I think the mortality is always going to be different.

Table 2 does try to describe the source of cases included. I think you need to separate the mortality of cases treated in the community from those treated in hospital. This would explain some of the significant variation in your mortality rates.

Risk factors: I thought this was a very informative section.

Discussion: From the countries included in the twelve studies what are the WHO reported numbers and mortality rates from these cases and how do they differ from the results in your meta-analysis and why?

6. PLOS authors have the option to publish the peer review history of their article (what does this mean?). If published, this will include your full peer review and any attached files.

Reviewer #1: **Yes: **Professor David A Watters

---

## [Author Response · Author response to Decision Letter 0]

30 Jul 2022

TO THE ACADEMIC EDITOR

We follow the recommendations of the link above for example We Used Level 1 heading for all major sections and Bold type, 18pt font; We cite figures as fig 1, fig 2, etc

2. Please include a caption for figures 5 to 11.

We included captions for figures 5 to 11

We included captions for supporting Information files at the end of the manuscript and update in-text citations

4. Thank you for submitting the above manuscript to PLOS ONE. During our internal evaluation of the manuscript, we found significant text overlap between your submission and the following previously published works, some of which you are an author.

- https://pubmed.ncbi.nlm.nih.gov/34238232/

- https://bmcinfectdis.biomedcentral.com/articles/10.1186/s12879-021-06369-0

- https://pubmed.ncbi.nlm.nih.gov/33075534/

- https://www.frontiersin.org/articles/10.3389/fphys.2021.665064/full

- https://link.springer.com/article/10.1007/s11255-020-02740-3?code=aa12583d-3699-4ff6-8f1f-78915123c553&error=cookies_not_supported

- https://bmcinfectdis.biomedcentral.com/articles/10.1186/s12879-021-06536-3

We changed the majority of sentences in the introduction and results section, we paraphrased sentences in discussions to avoid significant text overlap between our submission and the following previously published works. 

TO THE REVIEWER

Results: 

1.I believe Table 1 needs to have the time period of study, and perhaps a comment (in table 2 as to the vaccination status during the period of study.

We add the time period of study and the vaccination status during the period of study in table 2.

2. Next I am surprised by your mortality rate of 3%. I agree table 1 shows 43598 cases of Covid-19 from the relevant countries but I calculated from your percentage mortality figures and got 2555 deaths which is 5.8%. I think to clarify you need a column that includes the number of deaths as well as the mortality percent.

Thank you, dear reviewer, for making this calculation. Indeed, our mortality of 3% was wrong. And after reviewing in depth the mortality rates of all countries, we also found that there was a transcription error of the mortality rate values in the Jaspard et al study which was actually 5% instead of 29% (see revised table 1). And we added a column for the number of deaths as recommended and after recalculating the total number of deaths was 2103, recalculating the mortality rate is 4.8%. That's why I didn't calculate with the software anymore and removed the figure number 2 but calculated as you did to be more sure. Thank you very much for this

3. Table 1 should probably include also the source of the cohort of cases - eg community, hospital admitted and I think the mortality is always going to be different.

We have included in Table 1 the source of the cohort of cases. And indeed, the mortality was different depending on whether the case was hospitalized or community

4. Table 2 does try to describe the source of cases included. I think you need to separate the mortality of cases treated in the community from those treated in hospital. This would explain some of the significant variation in your mortality rates.

Table 2 describes the source of cases included in the column of setting

5. Risk factors: I thought this was a very informative section.

Thank you

Discussion: 

6. From the countries included in the twelve studies what are the WHO reported numbers and mortality rates from these cases and how do they differ from the results in your meta-analysis and why?

We compared the mortality obtained in our included studies and the mortality of the countries from which these studies came according to WHO, and we found huge differences in countries where the cases were only hospitalized but not a big difference in countries where community cases were also included. 

Below you can see the paragraph we added in the discussion:

The mortality prevalence of 4.8% in our study is higher than the mortality reported by the WHO. The reason for this is that the majority of the included patients in our meta-analysis were hospitalized, and in some included studies, the proportion of severe and critical patients was very high. The emergence of variants of concern may also influence case numbers and mortality, although almost all of the included studies were conducted in the first wave when these variants were not yet formally reported. Patient mortality in the DRC ranged from 13-29% versus 1.5% according to WHO data. In Ghana, mortality was 16% versus 0.87% according to WHO data. In Nigeria, mortality ranged from 3.3 to 9% versus 1.2% according to WHO data. In these countries, our meta-analysis shows high prevalence because most of the included patients are hospitalized, with a high proportion of severe cases. Mortality in South Africa and Ethiopia was close to WHO data (0.8 and 2.8 versus 2.5 and 1.5) because the studies included in these two countries also included cases from the community [1]

---

## [Decision Letter · Decision Letter 1]

25 Aug 2022

PONE-D-21-36467R1Risk factors for in-hospital mortality in COVID-19 patients in sub-Saharan Africa: A systematic review and meta-analysisPLOS ONE

Dear Dr. Bepouka,

Thank you for submitting your manuscript to PLOS ONE. After careful consideration, we feel that it has merit but does not fully meet PLOS ONE’s publication criteria as it currently stands. Therefore, we invite you to submit a revised version of the manuscript that addresses the points raised during the review process.

We look forward to receiving your revised manuscript.

Kind regards,

Alejandro Piscoya

Academic Editor

PLOS ONE

Journal Requirements:

Reviewers' comments:

Reviewer's Responses to Questions

**Comments to the Author**

1. If the authors have adequately addressed your comments raised in a previous round of review and you feel that this manuscript is now acceptable for publication, you may indicate that here to bypass the “Comments to the Author” section, enter your conflict of interest statement in the “Confidential to Editor” section, and submit your "Accept" recommendation.

Reviewer #1: All comments have been addressed

Reviewer #2: All comments have been addressed

2. Is the manuscript technically sound, and do the data support the conclusions?

Reviewer #1: Partly

Reviewer #2: Yes

3. Has the statistical analysis been performed appropriately and rigorously? 

Reviewer #1: Yes

Reviewer #2: I Don't Know

4. Have the authors made all data underlying the findings in their manuscript fully available?

Reviewer #1: Yes

Reviewer #2: Yes

5. Is the manuscript presented in an intelligible fashion and written in standard English?

Reviewer #1: (No Response)

Reviewer #2: Yes

6. Review Comments to the Author

Reviewer #1: The paper is improved and you have answered the comments from my earlier review other than a clearer calculation of the hospitalised and community mortality rates (see below).

Study selection P14: There is an error in your numbers for the cases found and excluded: "found 34263 articles. After eligibility criteria, 34491 were excluded" - you have excluded more articles than you found. There is a miscalculation here.

Table 2: Given none of the studies reported vaccination status, I suggest you change the table title to risk factors and leave out the vaccination status from the actual table (originally, I thought you may have a mix of vac status but all studies did not provide this). I don't think the vaccination status column is relevant given you have stated in the text and could do in a table footnote that none of the studies reported vaccination status. This can then be discussed as it would become increasingly relevant now that vaccinations are available.

I realise that some of the papers included both hospitalised and community patients and perhaps did not state what the mortality of each group was separately.

I still think it would be worth in the text stating what the mean mortality and range was of the patients who were hospitalised and, if possible, those that were known only to be treated or diagnosed in community but did not require admission. You will need to leave some studies out of this calculation but your hospitalised patient only studies suggest the mortality ranges from 4-29% which means hospitalisation = much higher mortality rate than 4.8%. This would be worth commenting on.

Table 1: I make the mortality rate from your hospitalised only studies 7.3% (439/5972) but some of the combined studies may also have stated the mortality rates for hospitalised and community patients separately.

I would suggest reorganising table 1 to show all the hospitalised only studies separately with totals compared with the combined studies.

Reviewer #2: Dear Author

I have checked the manuscript and you have addressed all the queries raised by the previous reviewers satisfactorily. So I have no comment to offer

bye

7. PLOS authors have the option to publish the peer review history of their article (what does this mean?). If published, this will include your full peer review and any attached files.

Reviewer #1: No

Reviewer #2: No

---

## [Author Response · Author response to Decision Letter 1]

31 Aug 2022

Dear academic editor and reviewer, thank you very much for your comments and corrections which help to improve the quality of our manuscript.

To the reviewer:

Find the different corrections we could make according to your remarks:

Study selection P14: There is an error in your numbers for the cases found and excluded: "found 34263 articles. After eligibility criteria, 34491 were excluded" - you have excluded more articles than you found. There is a miscalculation here. 

it is true that there was an error in the calculation, the corrected sentence is:

In the PRISMA flowchart, we can see the approach used for the document search. Initially, we found 34263 articles. After eligibility criteria, 34251 were excluded. Of the excluded studies, 5010 articles were recorded after duplicates were removed, 4890 were perceived as unrelated searches, and 108 articles met the exclusion criteria. Finally, 12 published articles were included (Fig 1). 

Table 2: Given none of the studies reported vaccination status, I suggest you change the table title to risk factors and leave out the vaccination status from the actual table (originally, I thought you may have a mix of vac status but all studies did not provide this). I don't think the vaccination status column is relevant given you have stated in the text and could do in a table footnote that none of the studies reported vaccination status. This can then be discussed as it would become increasingly relevant now that vaccinations are available.

We removed the vaccination status from the title and the table and we have placed a table footnote that none of the studies reported vaccination status. And we discussed this in the discussion like this:

 None of the studies reported vaccination status. The included studies were conducted in the first wave when vaccination was not yet effective. Vaccination could help reduce this high inpatient mortality, especially if vaccination coverage was high. In the literature, there is evidence of the efficacy of COVID-19 vaccines against severe forms and a reduction in mortality. Although most doses of COVID-19 vaccines have been administered in high and middle-income countries. The efficacy of the vaccines has been demonstrated in several countries where, for example, the efficacy of the two doses was 92%, 96%, and 89%, respectively, in Israel, the USA, and the UK [50]. But in Africa, up to the end of 2021, vaccination coverage was only 17% [51]. 

I realise that some of the papers included both hospitalised and community patients and perhaps did not state what the mortality of each group was separately.

I still think it would be worth in the text stating what the mean mortality and range was of the patients who were hospitalised and, if possible, those that were known only to be treated or diagnosed in community but did not require admission. You will need to leave some studies out of this calculation but your hospitalised patient only studies suggest the mortality ranges from 4-29% which means hospitalisation = much higher mortality rate than 4.8%. This would be worth commenting on.

Dear Reviewer, you are correct; the articles that included both inpatients and community patients did not report the mortality of each group separately.

In the text we stated: 

The range of inpatient mortality varied from 4% to 29%, with the mean mortality of the patients who were hospitalized of 7.3% (439/5972). The range of mortality for the combined studies (combining inpatients and community-diagnosed patients) varied from 0.8 to 9%, with a mean mortality rate of 4.4% (1664/37626). Separate data on the mean and range of mortality of patients treated or diagnosed in the community but not requiring admission were not clearly available in the included studies.

In the discussion, we added the sentence:

In addition, considering overall mortality was 4.8%; The range of inpatient mortality was 4–29% with a mean inpatient mortality of 7.3% (439/5972). Considering the range of mortality being 4–29%, one would expect higher mortality than 4.8%. This value of 4.8% is due to the fact that it is overall mortality combining inpatients and community patients, this value of 4.8% is due to the fact that it is the overall mortality combining inpatients and community patients, whose number is not negligible.

Table 1: I make the mortality rate from your hospitalised only studies 7.3% (439/5972) but some of the combined studies may also have stated the mortality rates for hospitalised and community patients separately.

I would suggest reorganising table 1 to show all the hospitalised only studies separately with totals compared with the combined studies.

We reorganized Table 1 by presenting all inpatient-only studies separately, comparing the totals with the combined studies.

To the academic editor:

the references added to the discussion section are the following:

50. Tregoning JS, Flight KE, Higham SL, Wang Z, Pierce BF. Progress of the COVID-19 vaccine effort: viruses, vaccines and variants versus efficacy, effectiveness and escape. Nat Rev Immunol. 2021 Oct;21(10):626-636. doi: 10.1038/s41577-021-00592-1. Epub 2021 Aug 9. PMID: 34373623; PMCID: PMC8351583.

51. WHO Africa faces 470 million COVID-19 vaccine shortfall in 2021. 2021. https://www.afro.who.int/news/africa-faces-470-million-covid-19-vaccine-shortfall-2021. Accessed August 28, 2022.

---

## [Editor Report · Decision Letter 2]

27 Sep 2022

Risk factors for mortality in COVID-19 patients in sub-Saharan Africa: A systematic review and meta-analysis

PONE-D-21-36467R2

Dear Dr. Bepouka,

We’re pleased to inform you that your manuscript has been judged scientifically suitable for publication and will be formally accepted for publication once it meets all outstanding technical requirements.

Kind regards,

Alejandro Piscoya

Academic Editor

PLOS ONE
---

## [Editor Report · Acceptance letter]

7 Oct 2022

PONE-D-21-36467R2 

Risk factors for mortality in COVID-19 patients in sub-Saharan Africa: A systematic review and meta-analysis 

Dear Dr. Bepouka:

I'm pleased to inform you that your manuscript has been deemed suitable for publication in PLOS ONE. Congratulations! Your manuscript is now with our production department. 

Kind regards, 

on behalf of

Professor Alejandro Piscoya 

Academic Editor

PLOS ONE